# The Role of the Microbiome in Oropharyngeal Squamous Cell Carcinoma: A Systematic Review

**DOI:** 10.3390/jpm15090399

**Published:** 2025-08-28

**Authors:** Jérôme R. Lechien

**Affiliations:** 1Department of Surgery, UMONS Research Institute for Health Sciences and Technology, University of Mons (UMons), B7000 Mons, Belgium; jerome.lechien@umons.ac.be; Tel.: +32-65373556; 2Department of Otolaryngology Head Neck Surgery, CHU Saint-Pierre, B1000 Brussels, Belgium; 3Department of Otolaryngology, Elsan Hospital, F75008 Paris, France

**Keywords:** oropharyngeal, oropharynx, cancer, carcinoma, microbiome, microbiota, bacteria, surgery, larynx, oncological, outcome, review

## Abstract

**Objective**: This systematic review aimed to investigate existing evidence regarding the implications of the microbiome in the initiation and progression of oropharyngeal squamous cell carcinoma (OPSCC). **Methods**: PubMed, Scopus, and Cochrane Library systematic searches were conducted according to the PRISMA statements to identify the relevant studies examining microbiome signatures, underlying molecular mechanisms, and their associations with clinical and oncological outcomes in OPSCC. The bias analysis was conducted with the MINORS. **Results**: Of the 83 identified papers, 12 met the inclusion criteria (298 OPSCC patients). *Spirochaetes* and most *Bacteroidetes* may be predominant in OPSCC versus control specimens, while *Proteobacteria* may be predominant in control tissues compared to tumor. *Leptotrichia*, *Selenomonas*, and *Treponema* trended to be overrepresented in OPSCC compared to control specimens. *Neisseria*, *Porphyromonas*, *Rothia*, *Streptococcus*, and *Veillonela* were predominantly reported in normal compared to OPSCC patient specimens. Microbiome compositional shifts were associated with chemoradiation response, HPV status, and addictions. Methodological heterogeneity was noted in sampling protocols, control selection, and analytical approaches, with limited statistical power due to small cohort sizes. **Conclusions**: OPSCC demonstrates different microbiome signatures from healthy tissues, influenced by HPV status and addictions. A microbiome shift is plausible from pre- to post-chemoradiotherapy, with the baseline microbiome acting as a predictive response factor; however, the low number of studies and substantial methodological heterogeneity across investigations limit the drawing of valid conclusions. The identification of key species is important in the development of OPSCC for developing personalized medicine considering bacterial mediators in terms of prevention, and targeted therapy using the microbiome–tumor–host interaction pathways.

## 1. Introduction

Head and neck squamous cell carcinoma (HNSCC) is the 6th most common adult cancer worldwide, accounting for 5.3% of all cancers [1,2]. Global HNSCC incidence has decreased significantly over the past three decades, primarily attributed to reduced tobacco and alcohol consumption [1,2]. Among HNSCC, the incidence of oropharyngeal squamous cell carcinoma (OPSCC) has progressively increased with annual incidence rates rising by 2.7% and 0.5% in males and females in the United States [3]. The rising incidence of OPSCC can be attributed to human papillomavirus (HPV) infection, which increased in the past few decades [4]. While HPV-associated OPSCC has become predominant in the United States, it is noteworthy that in many regions outside the US, particularly where anti-tobacco campaigns have been less successful or implemented more recently, tobacco and alcohol-related OPSCC continues to predominate, with significantly lower proportions of HPV-induced cases in countries across Europe, Asia, and South America [5]. The mechanisms underlying the interactions between HPV and the host environment in the OPSCC development are incompletely elucidated [4]. The number of studies dedicated to the role of microbial communities in respiratory and gastrointestinal pathophysiology increased over the past decade regarding the development of culture-independent metagenomic techniques [6]. In OPSCC, research has identified differential microbial signatures between HPV+, HPV− malignant, and healthy tissues, suggesting that dysbiosis-associated alterations influence tumor-promoting inflammation, carcinogenic metabolism, and chemoradiotherapy response patterns [7].

This systematic review aimed to investigate existing evidence regarding the implications of the microbiome in OPSCC tumorigenesis, progression, clinical phenotypes, and treatment-related endpoints.

## 2. Materials and Methods

Two researchers conducted the review using the Preferred Reporting Items for a Systematic Review and Meta-analysis (PRISMA) checklist [8]. Note that the protocol of review was not registered. Study selection criteria were defined with the PICOTS (Population, Intervention, Comparison, Outcome, Timing, and Setting) framework [9].

**Studies**: Studies published in peer-reviewed English-language journals between January 2000 and January 2025 were considered, including prospective and retrospective cohorts, cross-sectional investigations of cancer registries, and controlled trials exploring microbiome findings in OPSCC. Case reports, conference papers, preprints, and experimental animal studies were excluded.

**Participants and inclusion criteria**: Eligible studies provided OPSCC-specific patient data. Microbiome assessments derived from oropharyngeal or oral fluid or tumor specimens were considered eligible. There was no selection criteria based on the treatments, microbiome characterization methods, or demographic factors.

**Outcomes**: The primary outcomes consisted of the diversity and composition of the microbiome (taxonomic classification: phylum, class, order, family, genus, and species) and their associations with OPSCC. Secondary outcomes included study features (design, evidence-based level), demographics (mean/median age, sex ratio), oncological findings (grade, cTNM staging, treatments), and methodological approaches for microbiome evaluations (DNA extraction, amplification, quantification, and sequencing).

**Intervention and comparison**: There was no criterion for intervention. The data related to the type of treatments (surgery, chemo/radiotherapy) were extracted in studies investigating the prognostic value of microbiome findings.

**Timing and Setting**: There were no criteria for specific timing in the disease process.

### 2.1. Search Strategy

A University librarian and the author of the paper independently conducted the systematic literature search using PubMed, Scopus, and Cochrane Library databases. The following keywords were used for the literature research: Oropharynx; Oropharyngeal; Cancer; Squamous Cell Carcinoma; Oncological; Microbiome; Microbiota; Bacteria; and Outcomes. The investigators considered research reporting database abstracts, available full-texts, or titles with the search terms. The reference list of some articles, particularly reviews or meta-analyses, were considered for additional valuable studies. The studies were evaluated for the number of subjects, study design, inclusion and exclusion criteria, quality of trial, evidence-based level [10], demographics, and outcomes. Cohort studies from the same research team were carefully investigated for potential overlaps. Ethics committee approval was not required.

### 2.2. Bias Analysis

The bias analysis was conducted using the methodological index for non-randomized studies (MINORS), a validated tool for assessing study quality [11]. MINORS evaluates key methodological aspects on a scale of 0 (absent), 1 (inadequate/partial), or 2 (adequate). Items include study aim clarity, consecutive patient inclusion, prospective data collection, endpoint appropriateness, adequate follow-up period (for predictive studies), and acceptable lost-to-follow-up rates (<5%). For prospective studies, sample size calculation was also evaluated. The optimal MINORS score is 16 for non-comparative studies and 24 for comparative studies [11].

## 3. Results

Twelve studies were included, accounting for 298 OPSCC patients (Figure 1) [12,13,14,15,16,17,18,19,20,21,22,23]. There were 8 prospective controlled studies (evidence-based level (EBL) = 3C) and 4 uncontrolled/cross-sectional studies (EBL = 4) (Table 1). Six studies were excluded because authors reported data on oral/hypopharyngeal and oropharyngeal squamous cell carcinomas in a single group without specifically differentiating the anatomical locations [24,25,26,27,28,29]. There was no participant overlap in the included studies.

### 3.1. Demographics, Patients, and Tumor Stages

The data of 298 OPSCC patients were reported (Table 1, Figure 1). The gender ratio was specified in 7 studies [13,15,16,20,21,22,23]. There were 33 females (17.8%) and 152 males (82.2%), respectively. The mean age of patients ranged from 57.7 to 66.9 years (Table 1). The OPSCC stage, HPV status, and anatomical location information are described in Table 2. The cTNM information was reported in 5 studies [14,18,19,20,21]. The microbiome findings were investigated in cT1-T2 (n = 80), and cT3-T4 (n = 23) OPSCC patients. The majority of patients were cN1+ (n = 66; 64.7%). There were no data about patients with distant metastasis. The HPV status has been reported in 10 studies [12,14,15,16,17,18,19,20,21,22], accounting for 184 HPV+ (69.4%), 56 HPV− (21.1%), and 25 undetermined status (9.4%) patients. There was heterogeneity across studies for the HPV detection method, i.e., p16 immunostaining or PCR-based detection. The anatomical location was available in only four studies [13,18,22,23] and primarily consisted of tonsil OPSCC (Table 2). Two studies investigated the predictive value of the microbiome on the chemoradiation response [18,23].

### 3.2. Microbiome Outcomes

The microbiome outcomes are summarized in Table 1 and Table 3. The microbiome analyses were carried out on the following material: oral saliva (n = 7) [12,17,18,19,21,22,23], tumor/non-tumor tissue (n = 8) [13,14,15,16,18,20,22,23], and the stool (n = 1) [18]. Details about the methodology of microbiome analyses (e.g., DNA extraction, amplification, quantification, sequencing) are provided in Table 4. Among controlled studies, authors used the following material as controls: the saliva of healthy individuals [12,17,19,21], tissues of patients with obstructive sleep apnea syndrome [15,20], tissues of healthy individuals [13], or normal OPSCC-adjacent tissue [14]. De Martin et al. did not report significant differences in terms of α and β-diversities between OPSCC, NAT/CT [15].

Table 3 summarizes the statistically significant findings reported in the studies. Most authors used standardized statistical approaches, such as PERMANONA. Considering phylum transversal observations, *Spirochaetes* and most *Bacteroidetes* may be predominant in OPSCC versus control specimens, while *Proteobacteria* may be predominant in control tissues compared to tumor. No substantial trends can be extracted for other phyla due to controversial results across studies or lack of reported data.

The taxonomic profiling of at least two studies revealed that the following genera have been considered as predominant in OPSCC compared to control specimens: *Leptotrichia* [15,21,23], *Selenomonas* [15,21], and *Treponema* [14,21]. Moreover, *Neisseria* [14,21], *Porphyromonas* [14,15], *Rothia* [14,21], *Streptococcus* [14,21], and *Veillonela* [14,17,21] were predominantly reported in normal compared to OPSCC patient specimens.

*Actinomyces* was predominant in the salivary specimens of healthy controls in two studies [12,17], while being predominant in OPSCC tissue samples in two other studies [15,16]. Similar observations were found for *Dialister* [12,15], *Gemella* [15,18], *Tannarella* [13,15].

### 3.3. Influencing Factors on Microbiome Profile

The number of species detected in oral samples significantly decreased after chemoradiation [18], with controversial results about the α-diversity evolution from pre- to post-treatment [18,23]. Bahig et al. reported a potential predictive value of the microbiome on chemoradiation response, with higher baseline oral saliva and tumor tissue α-diversity in complete responders versus partial responders [23].

The potential influence of HPV infection on the microbiome profile has been primarily investigated in two studies [12,22]. Zakrzewski et al. observed higher microbial diversity in HPV− compared to HPV+ patients, while they did not report group differences in the salivary microbiome [22]. Bornigen et al. reported that some phyla and genera (e.g., *Actinomyces*, *Granulicatella*, *Campylobacter*, *Oribacterium*, *Rothia*, *Haemophilus parainfluenzae*, *Veillonella dispar*) were predominant in HPV+ versus HPV− OPSCC, while others (e.g., *Streptococcus anginosus*, *Mycoplasma*, *Peptoniphilus*) were predominant in HPV− OPSCC. Bornigen et al. [12] observed Rothia predominance in HPV-positive oral samples, whereas Zakrzewski et al. [22] reported a higher proportion of *Rothia* in HPV-negative samples compared to HPV-positive specimens. In the same vein, age, alcohol, and tobacco consumptions have been identified as factors influencing the microbiome profile [12,21] (Table 1).

### 3.4. Epidemiological Analysis

The mean MINORS was 12.0 ± 4.8, indicating substantial methodological limitations among the included studies (Table 5). There were no studies considering the inclusion of consecutive patients, while six teams clearly reported prospective data collection [17,18,20,21,22,23]. Endpoints (microbiome analyses) were potentially biased in two studies [13,14,15]. Precisely, Castaneda et al. assessed tobacco use, oral sexual behavior, alcohol consumption, and reflux; however, the distribution of these factors was asymmetrical between patient and control groups. Chan et al. analyzed the microbiome in surgical specimens without documenting pre- or perioperative antibiotic administration [14]. De Martin et al.’s study exhibited similar limitations regarding antibiotic documentation [15]. Study size calculation was provided in one study [12]. Most studies investigating microbiome patterns in OPSCC patients were limited by inadequate sample sizes, which limited subgroup analyses (HPV, alcohol, tobacco). In three studies, patient and control groups were matched for demographic, addiction, and infection findings [12,17,20]. Note that three studies lacked the specific documentation of alcohol and tobacco exposure [16,20,21,22].

The missing information related to the OPSCC stage [12,13,15,16,17,22,23], or HPV status [13,23] is an additional bias, limiting the finding interpretation. The selection of controls is an additional limitation of most studies with the consideration of obstructive sleep apnea syndrome tissues [15,20] and normal OPSCC-adjacent tissue [14] as healthy. Finally, some studies reported heterogeneity in the methods used for DNA extraction, amplification, quantification, and sequencing (Table 5).

## 4. Discussion

Multi-omic analysis and microbiome dynamics characterization are emerging in medicine and surgery due to the accessibility of metagenomic shotgun sequencing and microbiome functional analyses [30].

The present systematic review identified specific phyla/bacteria that may be significantly associated with the development/progression of OPSCC. Precisely, the transversal analysis of the literature demonstrated a predominance of *Spirochaetes* and *Bacteroidetes*, while *Proteobacteria* were predominant in control tissues compared to tumor specimens. Among the genera, *Leptotrichia*, *Selenomonas*, and *Treponema* showed higher representation in OPSCC compared to control specimens, whereas *Neisseria*, *Porphyromonas*, *Rothia*, *Streptococcus*, and *Veillonella* were more abundant in normal versus OPSCC tissue specimens.

The overrepresentation of *Bacteroidetes* in carcinoma tissues corroborates the findings found for laryngeal squamous cell carcinoma (LSCC) [31,32], with the detection of *Bacteroidetes* genera in approximately 15% of LSCC tissues. In oral squamous cell carcinomas, a recent review reported that *Bacteroidetes* was predominantly found in 13 of the 27 studies exploring microbiome features in oral squamous cell carcinoma [33]. *Bacteroidetes* was similarly involved in the development of gastrointestinal malignancies through multiple mechanistic pathways, including the modulation of WNT/β-catenin signaling, the activation of pro-inflammatory cytokine releases such as IL-8, and the upregulation of MAPK and WNT signaling cascades [34]. A better understanding of the mechanisms linking the tumor and *Bacteroidetes* may lead to the identification of transversal biomarkers across HNSCC. The relative abundance of *Bacteroidetes*, *Fusobacteria*, *Proteobacteria*, and *Actinobacteria* was inversely correlated with *Firmicutes* in LSCC [31,32]. In the present review, *Streptococcus (Firmicutes)* was consistently overrepresented in healthy tissues compared to OPSCC specimens, corroborating findings in LSCC and OSCC. Beyond its potential role as a biomarker for HNSCC, *Streptococcus* demonstrates prognostic value, as disease-free patients exhibited higher *Streptococcus* abundance in HNSCC [35]. Similarly to other phyla and genera, the mechanistic role of *Streptococcus* as protective genera in carcinogenesis remains largely unknown. In the oral cavity, the genus *Streptococcus* constitutes approximately 80% of the oral biofilm, where perturbations in oral streptococcal composition can lead to dysbiosis, altering host–pathogen interactions and resulting in oral inflammation [36]. The mechanistic relationship between decreased *Streptococcus* abundance in the upper aerodigestive tract mucosa and the development of chronic inflammation, related DNA damage, and carcinogenesis warrants further investigation. Similar observations can be made for *Rothia* genera that was transversally identified as abundant in healthy tissues compared to carcinoma specimens in HNSCC [31,33].

Despite the limited available literature, findings from the present review indicate potential distinct microbiome compositional shifts associated with chemoradiation response, HPV status, and tobacco-alcohol intoxications. Oliva et al. observed that the number of species detected in oral samples significantly decreased after chemoradiation [18], while Bahig et al. reported a potential predictive value of the microbiome on chemoradiation response, with higher baseline oral saliva and tumor tissue α-diversity in complete responders versus partial responders [23]. The predictive value of microbiome features in chemoradiotherapy response was investigated in studies considering all HNSCC sublocations [6,37]. Torozan et al. reported that patients with HNSCC exhibited significantly reduced alpha diversity compared to controls before and after chemoradiotherapy with an increase in the relative abundance of *Staphylococcus aureus* and *Escherichia coli* during chemoradiotherapy [37]. In a cohort of 52 HNSCC patients with stool samples, Hes et al. demonstrated that gut microbiome composition had predictive value for oral mucositis development, revealing a significant correlation between severe mucositis and reduced overall survival [6]. To date, the limited number of studies investigating oral or OPSCC microbiome dynamics during chemoradiotherapy and follow-up precludes definitive conclusions. However, such future investigations are important given the rising incidence of OPSCC and the substantial proportion of patients receiving chemoradiation therapy.

Some conflicting results across studies in the present review may be attributed to heterogeneity in inclusion criteria and confounding factors, which represent the primary limitation of the present review. Given the small number of patients, most studies did not investigate the impact of HPV status, laryngopharyngeal reflux disease, tobacco and alcohol consumption on the microbial composition and diversity, although these factors may influence the microbial composition [18,23,38]. In OSCC, alcohol consumption leads to chronic inflammation, dysbiosis, and an increased acetaldehyde level, leading to a tumor-promoting environment [28,39]. In LSCC, tobacco consumption was found to be significantly associated with the global community structure, specifically at lower taxonomic levels [28].

The potential heterogeneity across the studies in the inclusion criteria is an additional limitation. This limitation particularly concerns antibiotic exposure criteria at enrollment. Some authors did not exclude antibiotic use in the days preceding sampling or during surgery (sample collection time) [14,15], while others documented antibiotic consumption in the week before sample collection, which can significantly impact microbiome assessment [18]. Although most authors used saliva samples for microbiome analyses, some assessed the microbiome using tumor samples [13,14,15,16,20], which can limit comparisons across studies. The use of tissue from patients with identified diseases as controls (e.g., obstructive sleep apnea tonsils, tumor-adjacent tissue) represents a limitation due to potential dysbiosis related to the underlying disease. Finally, the full understanding of the role of the microbiome in the development of OSCC may require additional examinations, such as secretome and metaproteomic analyses, which provide the specific activity of species, while identifying the mediators influencing the specific activities. Spatial metagenomic and metaproteomic approaches represent another pathway for improvement, as they can determine the three-dimensional relationships among host tissues (tumor, peritumoral tissue) and microbial species. A summary of the key limitations of the literature and related considerations for future studies are provided in Table 6.

## 5. Conclusions

The current literature supports potential distinct microbiome signatures between OPSCC and non-cancerous tissues, with an overrepresentation of *Spirochaetes* and *Bacteroidetes* in OPSCC, and *Proteobacteria* and some Firmicutes (Streptococcus) in healthy tissues—features influenced by HPV. A microbiome shift is possible from pre- to post-chemoradiotherapy, with baseline microbiome acting as a predictive response factor; however, the low number of studies and substantial methodological heterogeneity across investigations limit the drawing of valid conclusions. The identification of key species is important in the development of OPSCC for developing personalized medicine considering bacterial mediators in terms of prevention, and targeted therapy using the microbiome–tumor–host interaction pathways.

## Figures and Tables

**Figure 1 jpm-15-00399-f001:**
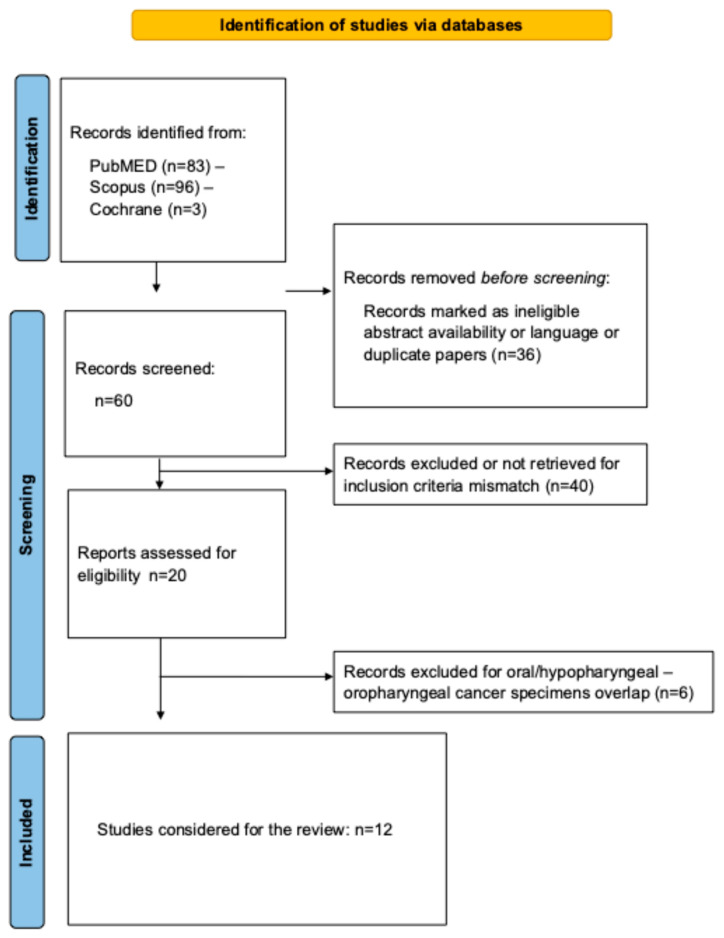
PRISMA flowchart.

**Table 1 jpm-15-00399-t001:** Demographic, clinical, and microbiome outcomes.

			Demographics			Other Considered				Primary
References	Design	EBL	N	F/M	Age (y)	Clinical Outcomes	CT Tissues	MS	Phyla Outcomes	Results
Bornigen	Prospective	3C	64 OPC	-	58.0	HPV, tobacco,	Saliva healthy	Or	Phylum and bacterial profile	OPC = OC
2017 [12]	Controlled		242 CT	54/188	-	alcohol, tooth status	controls		ACT, GRA, ORI, CAM, VED, ROT, HAP	HPV+ > HPV−
			52 OC	-	-				STA, PEP, MYC	HPV− > HPV+
									ROT, NEI, LAU	Smo− > Smo+
									LAC, BIF, ATO, PRE, STR, VEI	Smo+ > Smo−
Guerrero-Preston	Prospective	3C	11 OPC	3/8	62.0	cTN	Saliva healthy	Or	LAG/J, LAV, LAF, LSA, LRH	OPC > CT
2017 [19]	Controlled		25 CT				controls		STR	OC-OPC > CT
									HAE, NEI, LEP, LAU, AGG	CT > OPC-OC
Wolf	Prospective	3C	11 OPC	1/10	61.6	cN, HPV, tobacco	Saliva healthy	Or	Diversity; ACT, FIR (SCH, SEL), SPI (TRE)	OPC > CT
2017 [21]	Controlled		11 CT	1/10	47.7	alcohol, age, cT	controls		PRE, HAE, NEI, STR, VEI	CT > OPC
									cT, age, alcohol, tobacco association with MM	+
Hayes	Prospective	3C	12 OPC	-	-	HPV	Saliva healthy	Or	ACT, VED	OPC lower risk
2018 [17]	Controlled		254 CT	-	-		controls			
De Martin,	Prospective	3C	18 OPC	5/13	66.9	-	Healthy	Tu/ti	α/β-diversities	OPC = CT = NAT
2021 [15]	Controlled		14 CT	1/13	38.9		Tonsils of		SEL, LEP, ACT, MEG, NEI, GEM, PR6, CAP	OPC > CT
							OSAS patients		POR, FRE, DIA, FIL, TAN, ROT	CT > OPC
Oliva	UnProspective	4	22 OPC	-	-	HPV, tobacco,	-	Or	FUS, GEM, LEP, SEL	Stage III > Stage I-II
2021 [18]						stages, survival		St	Number of species (oral)	Pre > Post-CRT
								Tu	Oral α-diversity	Pre = post-CRT
Rajasekaran	Prospective	3C	46 OPC	7/39	57.7	cN, survival	Healthy	Tu	PMI, CHT, NEG, SHD, ORT (BAC, PAE)	cN+ > cN–
2021 [20]	Controlled		24 CT	5/19	46.4		Tonsils of		RHO, PRO, PEP	cN– > cN+
							OSAS patients		LACi, LLA	OPC > CT
Zakrzewski	UnProspective	4	13 OPC	1/12	62.0	HPV	-	Or	Tumor microbiome diversity	HPV− > HPV+
2021 [22]								Tu/ti	Saliva microbiome diversity	HPV+ = HPV−
									Tumor SPI, TRE/Richness	HPV+ > HPV−/CT > OPC
									Saliva LEP, saliva/tumor PRE, ROT	HPV− > HPV+
Bahig	UnProspective	4	9 OPC	0/9	62.0	CRT response	-	Or	Baseline α-diversity	Tumor = oral/CR > PR
2021 [23]						Tobacco		Tu	α-diversity, ACT, LEP	Pre > Post-CRT
									VEI, LEP	Tumor > oral
									Post-CRT VEI, ATO	PR > CR
Dhakal	Cross-sectional	4	53 OPC	9/44	58.0	-	-	Tu	ACT, SUL	OPC Abundance
2022 [16]									PSE, PSL	OPC scarcity
Chan	Prospective	3C	15 OPC	-	-	HPV, tobacco,	NAT	Tu/ti	FUS	OPC > CT
2022 [14]	Controlled		15 NAT	-	-	cT stage				
Castaneda	Prospective	3C	24 OPC	7/17	-	-	Healthy CT	Tu/ti	PRI, POG, TAF	OPC > CT
2023 [13]	Controlled		24 CT	7/17	-		Oral tissue			

The results reported in this table consisted of statistically significant differences/associations in microbiome patterns. Non-significant differences in microbiome signatures were not reported. ACT = Actinomyces; AGG = Aggregibacter; ATO = Atopoblum; BAC = Bacteroidaceae; BIF = Bifidobacterium; CAM = Campylobacter; CAP = Capnocytophaga; CHT = Chlamydia trachomatis; CR/PR = complete/partial response to chemoradiotherapy; CRT = chemoradiotherapy; CT = controls; DIA = Dialister; EBL = evidence-based level; FIL = Filifactor; FIR = firmicutes; FRE = Fretibacterium; GEM = Gemella; GRA = Granulicatella; HAE = haemophilus; HAP = Haemophilus parainfluenzae; LAC = Lactobacillus; LACi = Lactobacillus acidophilus; LAF = Lactobacillus fermentum; LAG/J = Lactobacillus gasseri/johnsonii; LAU = Lautropia; LAV = Lactobacillus vaginalis; LEP = Leptotrichia; LLA = Lactococcus lactis; LRH = Lactobacillus rhamnosus; LSA = Lactobacillus salivarius; MEG = Megasphaera; MS = microbiome sample; MYC = Mycoplasma; NAT = normal adjacent tissue; NEG = Neisseria gonorrhoeae; NEI = Neisseria; OC = oral carcinoma; OPC = oropharyngeal carcinoma; Or = oral; ORI = Oribacterium; ORT = Orientia tsutsugamushi; OSAS = obstructive sleep apnea syndrome; PAE = Paenibacillaceae; PEP = Peptoniphilus; PMI = Proteus mirabilis; POG = Porphyromonas gingivalis; POR = Porphyromonas; PR2/6/7 = prevotella_2/6/7; PRE = Prevotella; PRI = Prevotella intermedia; PRO = Propionibacteriaceae; PSL = pseudomonas libanensis; RHO = Rhodobacteraceae; ROM = Rothia mucilaginosa; ROT = Rothia; SCH = Schwartzia; SEL = Selenomonas_3; SHD = Shigella dysenteriae; Smo = smoker; St = stool; STA = Streptococcus anginosus; STE = Streptococcus; STR = streptococcus; SUL = Sulfurimonas; TAF = Tannerella forsythia; TAN = Tannarella; Tu/ti = tumor/healthy tissue; VED = Veillonella dispar; VEI = Veillonella.

**Table 2 jpm-15-00399-t002:** Oncological features.

																Anatomical Location					
		T Stage				N Stage							HPV Status				Oro-Hypo-	Tongue	Soft Palate/	OPC-	Unde-
References	N	cT1	cT2	cT3	cT4	N0	N1	N2a	N2b	N2c	N3a	M+	HPV+	HPV−	Unspecified	Tonsil	Pharynx	Base	OPC Wall	Larynx	Termined
Bornigen [12]	64	-	-	-	-	-	-	-	-	-	-	-	29	10	25	-	-	-	-	-	-
Castaneda [13]	24	-	-	-	-	-	-	-	-	-	-	-	-	-	-	4	1	2	11	2	4
Chan [14]	15	5	7	0	3	5	4	6			0	-	10	5	0	-	-	-	-	-	-
De Martin [15]	18	-	-	-	-	-	-	-	-	-	-	-	16	2	0	-	-	-	-	-	-
Dhakal [16]	53	-	-	-	-	-	-	-	-	-	-	-	38	15	0	-	-	-	-	-	-
Hayes [17]	12	-	-	-	-	-	-	-	-	-	-	-	5	7	0	-	-	-	-	-	-
Oliva [18]	22	5	5	6	5	3	12	2	1	1	2	0	22	0	0	12	0	7	1	0	0
Guerrero-Preston [19]	11	2	4	4	0	3	0	2	5	1	0	0	7	4	0	-	-	-	-	-	-
Rajasekaran [20]	46	46		0	0	23	23					0	46	0	0	-	-	-	-	-	-
Wolf [21] *	11	4	2	0	5	2	1	0	4	2	0	0	3	8	0	-	-	-	-	-	-
Zakrzewski [22] *	13	-	-	-	-	-	-	-	-	-	-	-	8	5	0	13	-	-	-	-	-
Bahig [23]	9	-	-	-	-	-	-	-	-	-	-	-	-	-	-	5	-	4	-	-	-
Total number	298	39	41	10	13	36	40	10	10	4	2	0	184	56	25	34	1	13	12	2	4

* p16 approach. Abbreviations: HPV = human papilloma virus; OPC = oropharyngeal carcinoma.

**Table 3 jpm-15-00399-t003:** Species.

	Bornigen			Castaneda		Chan		De Martin			Dhakal	Hayes		Oliva		Guerrero		Rajasekaran		Wolf		Zakrzewski		Bahig	
Phyla and Genera	OPC	OC	CT	OPC	CT	OPC	NAT	OPC	NAT	CT	OPC	OPC	CT	st I–II	st III	OPC	CT	OPC	CT	OPC	CT	OPC	CT	OPC	NAT
Bartonellaceae																		+++	+						
Burkholderiaceae																		+++	+						
Firmicutes								+++	+	+															
Catonella																				+	+++				
Dialister	+++	+++	+					+	+++	+++															
Filifactor								+	+++	+++															
Gemella								+++	+	+				+	+++										
Lactobacillales	+	+	+++																						
Lactobacillus vaginalis																+++	+	+++	+	+++	+				
Lactobacillus fermentum																+++	+	+++	+	+++	+				
Lactobacillus rhamnosus																+++	+	+++	+	+++	+				
Lactobacillus salivarius																+++	+	+++	+	+++	+				
Megasphaera								+++	+	+															
Moryella																				+++	+				
Parvimonas																						+	+++		
Selenomonas								+++	+	+				+	+++					+++	+				
Streptococcus						+	+++													+	+++				
Veillonella						+	+++					+	+++							+	+++			+++	+
Actinobacteria																									
Actinomyces	+	+	+++					+++	+	+	+++	+	+++							+++	+				
Bifidobacterium																				+++	+				
Rothia						+	+++	+++	+	+										+	+++				
Schwartzia																				+++	+				
Fusobacteriota																									
Fretibacterium								+	+++	+++															
Fusobacterium														+	+++							+++	+		
Leptotrichia								+++	+	+				+	+++					+++	+			+++	+
Bacteroidetes																									
Capnocytophaga								+++	+	+															
Flavobacterium						+++	+																		
Porphyromonas				+++	+	+	+++	+	+++	+++															
Prevotella_2								+	+	+															
Prevotella_6								+++	+	+															
Prevotella Intermedia				+++	+																				
Tannerella				+++	+			+	+++	+++															
Proteobacteria																									
Desulfovibrio																				+++	+				
Haemophilus																				+	+++				
Neisseria						+	+++	+++	+	+										+	+++				
Pasteurellaceae																				+	+++				
Pseudomonas											-														
Pseudomonas libanensis											-														
Sulfurimonas											+++														
Tenericutes																									
Mycoplasma						+++	+																		
Spirochaetes																									
Treponema						+++	+													+++	+				

The abundance of species was rated as mild (+), moderate (++), or high (+++). Abbreviations: CT = controls; OC = oral carcinoma; OPC = oropharyngeal carcinoma; NAT = normal adjacent tumor tissue.

**Table 4 jpm-15-00399-t004:** Microbiome analysis methods.

Reference	DNA Extraction Kit/Method	PCR Amplification Region	Quantification Method	Sequencing Platform
Bornigen [12]	QIAsymphony virus/bacteria Midi Kit	V4 16S rRNA (515F/806R)	qPCR	Illumina MiSeq v2
Dhakal [16]	TCGA data reanalysis	Not applicable	Not applicable	Not applicable
Hayes [17]	PowerSoil DNA Isolation Kit	V3–V4	DESeq2	454 FLX Titanium
Castaneda [13]	Phenol-chloroform method	3-deoxy-D-manno-octulosonic-acid gene	qPCR with Taqman probes	CFX 96 Real Time System BioRad
Chan [14]	Qiagen DNeasy Blood and Tissue Kit	V3–V4 16S rRNA (341F/806R)	Not specified	Illumina MiSeq
De Martin [15]	QIAamp DNA Stool Mini Kit with modifications	V4 16S rRNA	qPCR with KAPA HiFi polymerase	Illumina MiSeq v2
Oliva [18]	ZymoBIOMICS DNA/RNA Mini Kit	V4 16S rRNA	16S rRNA gene sequencing	Illumina MiSeq, NovaSeq 6000
Rajasekaran [20]	Not specified	Not specified	PathoChip array	Not specified
Guerrero-Preston [19]	Phenol/chloroform extraction	V3-V5 16S rRNA	454 pyrosequencing	Roche-454 FLX
Wolf [21]	QIAamp DNA Stool Mini Kit	V3-V5 16S rRNA	Qubit Thermo Fisher Scientific	Illumina MiSeq
Zakrzewski [22]	Qiagen QIAAmp Fast DNA Kit (modified)	V4 16S rRNA	16S rRNA gene sequencing	Illumina MiSeq
Bahig [23]	MO BIO PowerSoil DNA Isolation Kit	V4 16S rRNA	16S rRNA gene sequencing	Illumina MiSeq

**Table 5 jpm-15-00399-t005:** Bias analysis.

	Clearly	Inclusion of	Prospective	Endpoints	Unbiased	Follow-Up	<5% of	Study Size	Adequate	Contem-	Baseline	Adequate	Total
	Stated	Consecutive	Data	Appropriate	Endpoint	Adequate	Lost-to-	Prospective	Control	Porary	Group	Stat	MINORS
References	Aim	Patients	Collection	to Study	Assessment	Period	Follow-Up	Calculation	Group	Groups	Equivalence	Analyses	Score
Bornigen [12]	2	1	1	2	2	-	-	2	2	2	2	2	18
Castaneda [13]	2	0	1	1	1	-	-	0	1	1	1	1	9
Chan [14]	2	0	1	1	1	-	-	0	0	0	0	0	5
De Martin [15]	2	0	1	2	1			0	1	1	1	1	10
Dhakal [16]	2	0	0	2	1	-	-	0	-	-	-	-	5
Hayes [17]	2	1	2	2	2	2	-	0	2	2	2	2	19
Oliva [18]	2	1	2	2	2	2	-	0	0	0	0	2	13
Guerrero-Preston [19]	2	0	1	2	2	-	-	0	2	2	1	2	14
Rajasekaran [20]	2	0	2	2	2	2	-	0	2	2	2	2	18
Wolf [21]	2	0	2	2	2	-	-	0	2	1	1	2	14
Zakrzewski [22]	2	0	2	2	2	-	-	0	0	0	0	2	10
Bahig [23]	2	0	2	2	1	1	0	0	-	-	-	-	8

**Table 6 jpm-15-00399-t006:** Key findings for future studies.

Current Limitations	Recommendations for Future Studies
**Small sample sizes** limiting statistical power and subgroup analyses	1. Conducting multi-center collaborative studies with prospective sample
	size calculations based on specific primary outcomes.
	2. Establishing biobanks for OPSCC specimens to facilitate larger cohorts.
**Heterogeneity in inclusion/exclusion criteria** for antibiotic use	1. Standardizing antibiotic exclusion criteria (e.g., no antibiotics within 30 days before sampling).
	2. Documenting and reporting specific antibiotic exposure when unavoidable.
**Variable sample types** (saliva, tumor tissue, stool) limiting	1. Implementing multi-compartment sampling protocols (paired tumor/saliva samples).
cross-study comparisons	2. Establishing consensus guidelines for specimen collection, processing, and storage.
**Inadequate control groups** (using OSAS tissues or	1. Recruiting healthy controls matched for age, sex, diet, oral hygiene,
tumor-adjacent tissues as “normal”)	smoking, and alcohol consumption.
	2. Considering longitudinal sampling of at-risk individuals before cancer development.
**Inconsistent microbiome analysis methods** across studies	1. Adopting standardized protocols for DNA extraction, 16S rRNA
	amplification regions, and sequencing platforms.
	2. Establishing biobanks for OPSCC specimens to facilitate larger cohorts.
**Limited consideration of confounding factors** (HPV status, smoking,	1. Systematically documenting and stratifying analyses by HPV status,
alcohol, diet)	smoking pack-years, alcohol consumption patterns, oral hygiene.
**Predominance of cross-sectional designs** lacking temporal dynamics	1. Implementing longitudinal sampling at multiple timepoints (pre-treatment,
	during treatment, follow-up).
	2. Tracking microbiome changes during disease progression and treatment response.
**Focus on taxonomic profiling** with limited functional analysis	1. Integrating multi-omic approaches (metatranscriptomics, secretome,
	metaproteomics) to characterize functional activities of microbial communities.
**Limited investigation of host–microbiome interactions**	1. Including paired host tissue transcriptomics, immunoprofiling, and spatial
	characterization of microbiome relative to tumor and immune cell distribution.
**Inadequate reporting of clinical outcomes** in relation to microbiome	1. Correlating microbiome signatures with standardized clinical outcomes
	(treatment response, recurrence, survival) with adequate follow-up periods.
**Minimal validation of findings** across independent cohorts	1. Implementing discovery and validation cohort designs.
	2. Considering longitudinal sampling of at-risk individuals before cancer development.
**Lack of mechanistic insights** into how microbiome influences	1. Developing relevant in vitro and in vivo models to test causality of specific microbial signatures.
carcinogenesis or treatment response	2. Considering longitudinal sampling of at-risk individuals before cancer development.
**Limited spatial characterization** of microbiome within tumor	1. Implementing spatial metagenomics approaches to map microbiome
microenvironment	distribution within and around tumors.
**Inconsistent HPV detection methods** (p16 immunostaining vs. PCR)	1. Standardizing HPV detection with multiple complementary methods
	(DNA/RNA detection, genotyping, and p16 immunostaining).
**Insufficient reporting of oral hygiene status** and dental health	1. Documenting and controlling for oral hygiene practices, periodontal
	disease status, and dental health in all participants.

## Data Availability

The data presented in this study are available on request from the corresponding author.

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
