# Peer review of "The Role of the Microbiome in Oropharyngeal Squamous Cell Carcinoma: A Systematic Review"

_jpm, 2025, doi:10.3390/jpm15090399_

Round 1
Reviewer 1 Report
Comments and Suggestions for Authors
The submitted manuscript addresses a highly relevant and timely topic, with the potential to open new perspectives in the development of strategies for the prevention and treatment of oropharyngeal cancer. Unfortunately, the study’s findings and conclusions are rather limited and do not fully meet the stated objectives of the research.
While the author’s effort and intent are commendable, the existing literature on this subject is scarce and heterogeneous, which significantly limits the ability of the systematic review to generate clear conclusions or to outline well-defined future research directions.
An additional observation concerns the need for a more precise delineation of cancer topography, in accordance with internationally accepted classifications: lip and oral cavity, oropharynx, nasopharynx, hypopharynx, larynx, skin, etc. Within this topographical framework, it would be valuable to analyze potential differences in epidemiological data, risk factors, microbiological aspects, and so forth.
I am confident that a similar systematic review would be highly relevant in the coming years, provided that the medical community continues to investigate and publish more consistently and rigorously on this important subject.
Author Response
Reviewer #1: The submitted manuscript addresses a highly relevant and timely topic, with the potential to open new perspectives in the development of strategies for the prevention and treatment of oropharyngeal cancer. Unfortunately, the study’s findings and conclusions are rather limited and do not fully meet the stated objectives of the research. While the author’s effort and intent are commendable, the existing literature on this subject is scarce and heterogeneous, which significantly limits the ability of the systematic review to generate clear conclusions or to outline well-defined future research directions.
Indeed, this is the reason why a methodological analysis was provided, highlighting the methods used across studies for identifying the potential fields of improvements. The limitation paragraph was refined: p.16, line 60: “The potential heterogeneity across studies in the inclusion criteria is an additional limitation. This limitation particularly concerns antibiotic exposure criteria at enrollment. Some authors did not exclude antibiotic use in the days preceding sampling or during surgery (sample collection time) 14,15, while others documented antibiotic consumption in the week before sample collection, which can significantly impact microbiome assessment 18. Although most authors used saliva samples for microbiome analyses, some assessed the microbiome using tumor samples 13–16, 20, which can limit comparisons across studies. The use of tissue from patients with identified diseases as controls (e.g., obstructive sleep apnea tonsils, tumor-adjacent tissue) represents a limitation due to potential dysbiosis related to the underlying disease. Finally, the full understanding of the role of microbiome in the development of OSCC may require additional examinations, such as secretome and metaproteomic analyses, which provide the specific activity of species, while identifying the mediators influencing the specy activities. The spatial metagenomic and metaproteomic is another way of improvement, determining the 3D relationship across host tissues (tumor, peritumoral tissue) and species.”
An additional observation concerns the need for a more precise delineation of cancer topography, in accordance with internationally accepted classifications: lip and oral cavity, oropharynx, nasopharynx, hypopharynx, larynx, skin, etc. Within this topographical framework, it would be valuable to analyze potential differences in epidemiological data, risk factors, microbiological aspects, and so forth.I am confident that a similar systematic review would be highly relevant in the coming years, provided that the medical community continues to investigate and publish more consistently and rigorously on this important subject.
Thank you. This point supports the publication of the review identifying the current limitations. To highlight the importance of the reviewer’s analysis, an additional table was provided: see Table 6 with current literature limitations and potential solutions.
Reviewer 2 Report
Comments and Suggestions for Authors
This manuscript describes a systematic review on the role of the microbiome in the development, advancement, clinical characteristics, and treatment outcomes of oropharyngeal squamous cell carcinoma (OPSCC).
The subject is of great interest.
Here are my comments:
Abstract: The abstract is solid, but in my opinion can be improved a little.
Line 12: ‘tumorigenesis, progression, clinical phenotypes, and treatment-related endpoints’ reads a bit like a list. Perhaps a slight rephrasing could improve its flow.
Line 16-17: Consider if 'appear to be' and 'may be' understate the strength of the evidence in your review. A more confident formulation might be appropriate for the abstract.
Introduction: The introduction outlines the complex issues associated with this cancer. It offers sufficient background to contextualize the research question for the reader.
Lines 38-39: ‘The rising incidence of OPSCC can be attributed to human papillomavirus (HPV) infection, which increased in the past few decades’. I suggest adding a sentence to underline that outside the US, where campaigns against smoking have not been so successful, the rate of HPV induced OPSCC is much lower. I recommend to read and cite the following paper concerning the above mentioned aspect: Gallus R et al. Accuracy of p16 IHC in Classifying HPV-Driven OPSCC in Different Populations. Cancers . 2023;15. doi:10.3390/cancers15030656
Line 42: ‘led to an increase of the number of studies dedicated to’ is a bit wordy.
Materials and methods: The section is well-structured and thorough, the criteria for studies, participants, and specimens are clearly outlined.
Line 57: Consider specifying whether preprints or conference abstracts were excluded.
Outcomes: consider breaking it into shorter sentences for better readability. Consider defining what ‘EBL 10’ means also in the manuscript, not only in the table footnotes.
Line 79: I suggest replacing ‘have been’ with ‘were’ for consistency with past tense.
The paragraph of MINORS items could be slightly more concise.
Results: The content is rich and detailed, showing careful review of the studies.
Microbiome Outcomes
‘Table 3 summarizeS…’ If possible, include a brief mention of the types of statistical tests employed across the studies (e.g., ANOVA, t-tests).
‘No substantial trends can be extracted for other phyla.’ Does the author mean the results were statistically non-significant or just inconsistent?
Discussion: The paragraph is well-written, however it contains some redundant concepts, such as the oncogenic potential of Bacteroidetes and study limitations.
I would suggest to streamline repeated ideas to maintain focus.
Lines 31-35: How Streptococcus or Rothia abundance may affect tumor progression could be more critically explored.
Please also revise grammatical errors.
Conclusions: Line 72: The reference of 'both' is unclear.
Line 75: ‘the identification of key species... is important’ is a bit generic. I would strengthen this statement by specifying why identifying species is important.
References: The 41 bibliographic references are in general appropriate, one more has
been suggested above.
Figure 1: Identification: I would suggest indicating how many records were identified for each database. Please check the number of records removed.
I would also suggest adding the PRISMA checklist in the supplementary files.
Abbreviations: Please check the second one, line 97.
Author Response
Reviewer #2:
This manuscript describes a systematic review on the role of the microbiome in the development, advancement, clinical characteristics, and treatment outcomes of oropharyngeal squamous cell carcinoma (OPSCC). The subject is of great interest.
Thank you.
Here are my comments:
Abstract: The abstract is solid, but in my opinion can be improved a little.
Line 12: ‘tumorigenesis, progression, clinical phenotypes, and treatment-related endpoints’ reads a bit like a list. Perhaps a slight rephrasing could improve its flow.
Corrected to: “This systematic review aimed to investigate existing evidence regarding the implications of microbiome in the initiation and progression of oropharyngeal squamous cell carcinoma (OPSCC).”
Line 16-17: Consider if 'appear to be' and 'may be' understate the strength of the evidence in your review. A more confident formulation might be appropriate for the abstract.
Corrected in all manuscript sections: “Spirochaetes and most Bacteroidetes may be predominant in OPSCC versus control specimens, while Proteobacteria may be predominant in control tissues compared to tumor. Leptotrichia, Selenomonas, and Treponema trended to be overrepresented in OPSCC compared to control specimens.”
Introduction: The introduction outlines the complex issues associated with this cancer. It offers sufficient background to contextualize the research question for the reader.
Lines 38-39: ‘The rising incidence of OPSCC can be attributed to human papillomavirus (HPV) infection, which increased in the past few decades’. I suggest adding a sentence to underline that outside the US, where campaigns against smoking have not been so successful, the rate of HPV induced OPSCC is much lower. I recommend to read and cite the following paper concerning the above mentioned aspect: Gallus R et al. Accuracy of p16 IHC in Classifying HPV-Driven OPSCC in Different Populations. Cancers . 2023;15. doi:10.3390/cancers15030656.
Done: Introduction, p.2, line 39: “While HPV-associated OPSCC has become predominant in the United States, it is noteworthy that in many regions outside the US, particularly where anti-tobacco campaigns have been less successful or implemented more recently, tobacco and alcohol-related OPSCC continues to predominate, with significantly lower proportions of HPV-induced cases in countries across Europe, Asia, and South America.5”
Line 42: ‘led to an increase of the number of studies dedicated to’ is a bit wordy.
Rephrased: “The mechanisms underlying the interactions between HPV and the host environment in the OPSCC development are incompletely elucidated4. The number of studies dedicated to the role of microbial communities in respiratory and gastrointestinal pathophysiology increased over the past decade regarding the development of culture-independent metagenomic techniques”
Materials and methods: The section is well-structured and thorough, the criteria for studies, participants, and specimens are clearly outlined.
Line 57: Consider specifying whether preprints or conference abstracts were excluded.
Done: Method, p.3, line 57: “Case reports, conference paper, preprints, and experimental animal studies were excluded.”
Outcomes: consider breaking it into shorter sentences for better readability. Consider defining what ‘EBL 10’ means also in the manuscript, not only in the table footnotes.
Done: p.3, line 79: “The studies were evaluated for the number of subjects, study design, inclusion and exclusion criteria, quality of trial, evidence-based level10, demographics, and outcomes.”
Line 79: I suggest replacing ‘have been’ with ‘were’ for consistency with past tense.
Done.
The paragraph of MINORS items could be slightly more concise.
Done: p.3, line 84: “The bias analysis was conducted using the methodological index for non-randomized studies (MINORS), a validated tool for assessing study quality.11 MINORS evaluates key methodological aspects on a scale of 0 (absent), 1 (inadequate/partial), or 2 (adequate). Items include study aim clarity, consecutive patient inclusion, prospective data collection, endpoint appropriateness, adequate follow-up period (for predictive studies), and acceptable lost-to-follow-up rates (<5%). For prospective studies, sample size calculation was also evaluated. The optimal MINORS score is 16 for non-comparative studies and 24 for comparative studies.11”
Results: The content is rich and detailed, showing careful review of the studies.
Microbiome Outcomes
‘Table 3 summarizeS…’ If possible, include a brief mention of the types of statistical tests employed across the studies (e.g., ANOVA, t-tests).
Corrected. We do not use ANOVA or statistical approach because the heterogeneity of studies and the related inability to do meta-analysis. Among studies, they used PERMANOVA as recommended for microbiome analyses.
We specified: “Most authors used standardized statistical approaches, such as PERMANONA.”
‘No substantial trends can be extracted for other phyla.’ Does the author mean the results were statistically non-significant or just inconsistent?
We specified: Results, p/8, line 14: “No substantial trends can be extracted for other phyla due to controversial results across studies or lack of reported data.”
Discussion: The paragraph is well-written, however it contains some redundant concepts, such as the oncogenic potential of Bacteroidetes and study limitations. I would suggest to streamline repeated ideas to maintain focus. Lines 31-35: How Streptococcus or Rothia abundance may affect tumor progression could be more critically explored.
Discussion was reworked (all changes in green in the revised version), reduced through the removal of redundance findings, to maintain focus. Moreever, the role of streptococcus was specified: “Similar to other phyla and genera, the mechanistic role of Streptococcus as protective genera in carcinogenesis remains largely unknown.”
Please also revise grammatical errors.
Done.
Conclusions: Line 72: The reference of 'both' is unclear.
It was modified: “The current literature supports potential distinct microbiome signatures between OPSCC and non-cancerous tissues, with overrepresentation of Spirochaetes and Bacteroidetes in OPSCC, and Proteobacteria and some Firmicutes (Streptococcus) in healthy tissues, these features influenced by HPV.”
Line 75: ‘the identification of key species... is important’ is a bit generic. I would strengthen this statement by specifying why identifying species is important.
Done: line 75 (+ conclusion of the paper): “The identification of key species is important in the development of OPSCC for developing personalized medicine considering bacterial mediators in terms of prevention, and targeted therapy using the microbiome-tumor-host interaction pathways.”
References: The 41 bibliographic references are in general appropriate, one more has
been suggested above.
The suggested reference was added accordingly.
Figure 1: Identification: I would suggest indicating how many records were identified for each database. Please check the number of records removed.
Done. Figure 1 was modified.
I would also suggest adding the PRISMA checklist in the supplementary files.
Done.
Abbreviations: Please check the second one, line 97.
Done. EBL was defined as its first use.
Reviewer 3 Report
Comments and Suggestions for Authors
This manuscript represents a systematic review exploring the connection between microbiome (bacteria species) and OPSCC. The study adopted a clear protocol and followed the classical PRISMA guidelines. The results, though limited by the number of studies (only 12, with 298 cases), are well-described and discussed. I have a few minor comments only.
- Was the protocol of the study registered, as suggested by PRISMA guidelines?
- The author should provide a list of excluded studies along with reasons for exclusion.
- What does MM mean on Table 1?
- By any chance, could the data be subjected to meta-analysis?
- Although not directly in the scope of the article, I think that, after visiting the current literature and highlighting the main issues of the studies, the author could come up with some orientations of how-to procedure with an ideal study, defending, for example, the best strategy, sample, procedure, statistical analysis, reporting and validation. This can give an interesting closing to the discussion.
Author Response
Reviewer #3:
This manuscript represents a systematic review exploring the connection between microbiome (bacteria species) and OPSCC. The study adopted a clear protocol and followed the classical PRISMA guidelines. The results, though limited by the number of studies (only 12, with 298 cases), are well-described and discussed. I have a few minor comments only.
- Was the protocol of the study registered, as suggested by PRISMA guidelines?
No. We mention to the local IRB, which reported that there is no need of registration.
We added in the methods, line 52: “Note that the protocol of review was not registred.”
- The author should provide a list of excluded studies along with reasons for exclusion.
Figure 1 was modified accordingly.
- What does MM mean on Table 1?
Microbiome sample. The abbreviation was changed into “MS” and specified in the footnotes.
- By any chance, could the data be subjected to meta-analysis?
Unfortunately no regarding the heterogeneity across studies. As recommended by another reviewer, Table 6 was added to highlight heterogeneity across studies and potential ways of improvements.
- Although not directly in the scope of the article, I think that, after visiting the current literature and highlighting the main issues of the studies, the author could come up with some orientations of how-to procedure with an ideal study, defending, for example, the best strategy, sample, procedure, statistical analysis, reporting and validation. This can give an interesting closing to the discussion.
Indeed. The table 6 was added for this purpose.
Round 2
Reviewer 1 Report
Comments and Suggestions for Authors
Thank you for the revised version. Wishing you success with your research in this field.